# Sputtered Platinum Thin-films for Oxygen Reduction in Gas Diffusion Electrodes: A Model System for Studies under Realistic Reaction Conditions

**Gustav W. Sievers** [1,2,†]**, Anders W. Jensen** [1,†] **, Volker Brüser** [2]**, Matthias Arenz** [3,*] **and María Escudero-Escribano** [1,*]

[1]  Nano-Science Centre, Department of Chemistry, University of Copenhagen, Universitetsparken 5, 2100 Copenhagen Ø, Denmark; sievers@inp-greifswald.de (G.W.S.); awj@chem.ku.dk (A.W.J.)
[2]  Leibniz Institute for Plasma Science and Technology, Felix-Hausdorff-Strasse 2, 17489 Greifswald, Germany; brueser@inp-greifswald.de
[3]  Department of Chemistry and Biochemistry, University of Bern, 3006 Bern, Switzerland
[*]  Correspondence: matthias.arenz@dcb.unibe.ch (M.A.); maria.escudero@chem.ku.dk (M.E.-E.); Tel.: +41-31631-5384 (M.A.); +45-35-32-83-90 (M.E.-E.)
[†]  These authors contributed equally to this work.

**Abstract:** The development of catalysts for the oxygen reduction reaction in low-temperature fuel cells depends on efficient and accurate electrochemical characterization methods. Currently, two primary techniques exist: rotating disk electrode (RDE) measurements in half-cells with liquid electrolyte and single cell tests with membrane electrode assemblies (MEAs). While the RDE technique allows for rapid catalyst benchmarking, it is limited to electrode potentials far from operating fuel cells. On the other hand, MEAs can provide direct performance data at realistic conditions but require specialized equipment and large quantities of catalyst, making them less ideal for early-stage development. Using sputtered platinum thin-film electrodes, we show that gas diffusion electrode (GDE) half-cells can be used as an intermediate platform for rapid benchmarking at fuel-cell relevant current densities (~1 A cm$^{-2}$). Furthermore, we demonstrate how different parameters (loading, electrolyte concentration, humidification, and Nafion membrane) influence the performance of unsupported platinum catalysts. The specific activity could be measured independent of the applied loading at potentials down to 0.80 V$_{RHE}$ reaching a value of 0.72 mA cm$^{-2}$ at 0.9 V$_{RHE}$ in the GDE. By comparison with RDE measurements and Pt/C measurements, we establish the importance of catalyst characterization under realistic reaction conditions.

**Keywords:** electrocatalysis; oxygen reduction; ORR; gas diffusion electrode; platinum; fuel cells; thin-films; benchmarking; mass transport

## 1. Introduction

Fuel cell technologies, which convert chemical energy directly into clean electricity, are expected to play a key role in environmentally friendly energy conversion schemes [1]. Polymer exchange membrane fuel cells (PEMFCs) are very promising both for transportation and stationary applications [2,3]. Platinum-based catalysts typically catalyze both the oxidation of hydrogen at the anode and the reduction of oxygen at the cathode of PEMFCs; as a result, the overall cost of the PEMFC technology is closely linked with the platinum loading [4]. The slow kinetics of the oxygen reduction reaction (ORR) causes significant losses in cell potential. Consequently, the loading at the cathode accounts for the majority of platinum usage in PEMFC. In order to reduce the Pt loading, most scientific and industrial

research focuses on developing new ORR catalysts that present both high activity and stability for the ORR [5–7].

The ORR activity can be enhanced by modifying the geometric [8] or electronic structure [9] of the Pt-based electrocatalytic surfaces. To improve the catalyst mass activity, we can enhance: i) the electrochemically active surface area (ECSA) [10] and/or ii) the specific catalytic activity by modifying the electronic properties of platinum by alloying [11–13]. As a result, numerous catalyst concepts have been developed with activities far exceeding commercial Pt catalysts using thin-film rotating disk electrode (RDE) measurements in liquid half cells [10]. However, the impressive activity enhancements obtained with RDE have not been translated to real devices [14]. To the best of our knowledge, only a single concept based on dealloyed bimetallic Pt catalysts have exceeded the Department of Energy (DOE) target of activity normalized by mass of precious group metal (PGM) of 0.44 A/mg$_{PGM}$ at 0.9 V$_{RHE}$ in a membrane electrode assembly (MEA) [15].

Benchmarking Pt-based catalysts using the RDE method allows a fast investigation of the trends in electrocatalytic activity of nanoparticles supported on high surface area carbon materials and sputtered thin films [16–18]. However, there are many differences between RDE and MEA measurements. These include different catalyst loadings; in RDE, typical 5–20 µg$_{Pt}$ cm$^{-2}$ are used; on the other hand, MEAs typical employs loadings in the range of 0.1-0.5 mg$_{Pt}$ cm$^{-2}$ [19]. Secondly, different testing parameters are utilized. In RDE measurements, the potential is typically swept with a rate between 5 and 100 mV s$^{-1}$ at room temperature [17]. In contrast, MEA testing is generally carried-out at elevated temperatures with constant currents or potentials [20]. A third difference is the reaction environment at the point of the active site. In MEAs, the oxygen reduction occurs at the triple phase boundary where O$_2$ gas, proton transport mechanisms and electrical conductive particles coexist [21]. Conversely, the catalytic layer in the RDE configuration is immersed in an oxygen-saturated aqueous electrolyte, enabling superior access of oxygen, protons and electrons [22].

Another drawback of RDE benchmarking is the low diffusion and solubility of molecular oxygen in aqueous electrolytes. Consequently, the rate of the reaction is under full mass transport limitation at current densities orders of magnitude lower than actual fuel cells. Using the RDE technique, it is therefore, only possible to measure activities at low overpotentials (typically 0.9 V *versus* the reversible hydrogen electrode, V$_{RHE}$), far from the working potential in real fuel cells (0.6–0.8 V$_{RHE}$). While trends in ORR rate can be investigated at wider potential ranges with use of the Koutecky-Levich analysis [23], it is associated with great uncertainties in weakly adsorbing electrolytes. In MEAs, the catalyst layers are applied on porous gas diffusion electrodes in order to enable diffusion of O$_2$ from the gas phase. However, MEA benchmarking requires large quantities of catalyst and specialized equipment; as a result, it is less accessible for early stage catalysts development.

Since Zatilis et al. showed that superior, near mass transport free current could be achieved with a floating electrode configuration using an artificial gas diffusion interface [24], several gas diffusion electrode (GDE) half-cells designs have been reported for ORR benchmarking [25–28]. These works include measurements in concentrated phosphoric acid at elevated temperature [25], methanol and ethanol oxidation [26] and Pt/C in HClO$_4$ showing comparable activity to MEA, both at room temperature [27] and at elevated temperatures using a Nafion separating membrane [28]. Furthermore, different GDE setups have recently been used for both CO [29] and CO$_2$ reduction [30–32].

In this study, we utilize a GDE setup that enables testing commercial gas diffusion layers with an interchangeable Nafion membrane resembling actual PEMFCs configuration [28]. Using sputtered platinum as a model catalyst, we demonstrate the influence of different testing parameters (electrolyte, scan rate, membrane and humidification) on the performance of the unsupported catalysts. By investigating the effect of different catalyst loadings in both GDE and RDE, we identify key design principles and establish the importance of benchmarking under realistic reaction conditions.

## 2. Materials and Methods

### 2.1. Preparation of Pt Thin-Film Electrodes

For the Pt thin film deposition, the magnetron electrode was equipped with a planar target of 99.95 % Pt (Junker Edelmetalle, Waldbüttelbrunn, Germany). It was located at the superior part of the recipient. The RF generator (Advanced Energy, Fort Collins, CO, USA) had a driving frequency of 13.56 MHz. The reactor chamber was evacuated to a base pressure of $5 \times 10^{-3}$ Pa. An argon plasma was ignited in the chamber at a working pressure of 5 Pa. The platinum loadings of the sputtered Pt TFs were estimated by inductively coupled plasma mass spectrometry (ICP-MS) measurements (Aurora Elite, Bruker, Billerica, MA, USA). For this purpose, the catalysts were digested in aqua regia freshly mixed with 30% HCl (Suprapur, Merck, Darmstadt, Germany) and 65% $HNO_3$ (Suprapur, Merck, Darmstadt, Germany) in a volumetric ratio of 3:1, respectively. A Dektak 3ST profilometer (Veeco, Plainview, NY, USA) was used to measure the film thickness. Corresponding to the Pt loadings of 11.7 $\mu g_{Pt}$ cm$^{-2}$, 16.2 $\mu g_{Pt}$ cm$^{-2}$, 29.4 $\mu g_{Pt}$ cm$^{-2}$ and 66.2 $\mu g_{Pt}$ cm$^{-2}$, the films had a thickness of 15 nm, 20 nm, 37 nm and 83 nm, respectively.

### 2.2. Physical Characterization

The morphology of the sputtered Pt thin-film was investigated with scanning electron microscopy (JSM 7500F, JEOL, Tokyo, Japan) with a field-emission gun, a semi-in-lens conical objective lens and a secondary electron in-lens detector for high-resolution and high-quality image observation of structural features of the deposited films at a maximum specified resolution of 1.0 nm at 15 keV. The technique enables imaging the surface without any preparative coatings. The GDE was directly placed onto the SEM unit holder. The phase identity and crystallite size of the Pt catalyst was investigated by means of x-ray diffraction (XRD) using a Bruker D8 Advance Diffractometer (Bruker, Billerica, MA, USA), with measurements performed over a 2Theta range from 20° to 80°, step width 0.05° and 5 s per step. Cu was used as the anode target.

### 2.3. Chemicals, Materials and Gases

Deionized ultrapure water (resistivity >18.2 MΩ·cm, total organic carbon (TOC) < 4 ppb) from a Milli-Q system (Millipore, Burlington, MA, USA) was used for acid dilutions, catalyst ink formulation, and the GDE cell cleaning. Perchloric acid (70% HClO4, Suprapur, Merck) were used for electrolyte preparation. The Nafion membrane (Nafion 117, 183 μm thick, Chemours, Wilmington, DE, USA) and the gas diffusion layer (GDL) with a microporous layer (MPL) (H23 C2, Freudenberg, Weinheim, Germany) were employed in the GDE cell measurements. The following gases (Air Liquide, Taastrup, Denmark) were used in electrochemical measurements: Ar (99.999%), O2 (99.999%), and CO (99.97%).

### 2.4. Gas Diffusion Electrode (GDE)

Electrochemical measurements were carried out using a GDE setup [28]. In contrast to RDE, benchmarking is limited to low overpotentials, the GDE setup allow testing of ORR activity at high current densities under realistic mass transport conditions. The GDE half-cell consists of two cell components (see Figure 1): (i) a lower component made of body of stainless steel with a flow field and gas supply (ii) an upper cell body of polytetrafluoroethylene (PTFE) with electrolyte and counter electrode (platinum mesh) and reference electrode (reversible hydrogen electrode, RHE). For cleaning, the Teflon upper part was soaked in mixed acid ($H_2SO_4$:$HNO_3$ = 1:1, v:v) overnight. Subsequently, it was rinsed thoroughly by ultrapure water, and boiled three times.

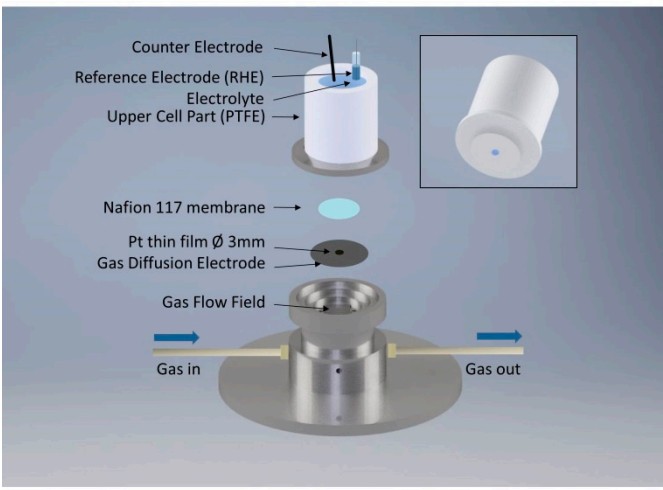

**Figure 1.** Schematic illustration of the gas diffusion electrode (GDE) half-cell setup, showing the catalyst layer and membrane sandwiched between the upper PTFE with electrolyte and the lower stainless steel body with the gas flow field.

The geometric surface area of the working electrode was 3 mm. The GDE was used with and without membrane between electrolyte and working electrode. When the membrane was used, it was pressed during assembling of the cell to the catalyst layer. For measurements using humidification, the gas was bubbled through a gas humidifier.

*2.5. Electrochemical Measurements*

All electrochemical measurements were performed using a computer-controlled potentiostat (ECi 200, NordicElectrochemistry, Copenhagen, Denmark). The measurements were performed using 4 M $HClO_4$ and 1 M $HClO_4$ aqueous solutions at room temperature. The high electrolyte concentrations reduce the solution resistance between working electrode, counter electrode and reference electrode. Prior to the measurements, the working electrode was purged from the backside (through the gas diffusion layer) with $O_2$ gas and the catalyst was conditioned by potential cycles between 0.1 and 1.0 $V_{RHE}$ at a scan rate of 100 mV s$^{-1}$ until a stable cyclic voltammogram could be observed (*ca*. 30 cycles). The ECSA of the catalyst was determined by conducting CO stripping voltammetry. The working electrode was held at 0.05 V during a CO purge through the GDL for 2 min followed by an Ar purge for 10 min. The ECSA was determined from the CO ($Q_{CO}$) oxidation charge recorded at a scan rate of 50 mV s$^{-1}$ and the respective Pt loading ($L_{Pt}$) using a fixed conversion coefficient of 390 μC cm$^{-2}$$_{Pt}$ [16]; ECSA = $Q_{CO}/(L_{Pt} \times 390$ μC cm$^{-2}$$_{Pt})$. To determine the ORR activity, linear sweep voltammetry (LSV, anodic scan) was conducted by purging the electrode with $O_2$ from below and scanning the potential at a scan rate of 50 mV s$^{-1}$ or 100 mV s$^{-1}$. The polarization curves were corrected for the non-faradaic background by subtracting the cyclic voltammograms (CVs) recorded in Ar-purged electrolyte at the identical scan rate. Furthermore, the resistance between the working and reference electrode (~10 Ω) was determined using an AC signal (5 kHz, 5 mV) and thereafter compensated for using analogue positive feedback scheme of the potentiostat. The resulting effective resistance was 1 Ω or less for each experiment.

## 3. Results and Discussion

*3.1. Structural Characterization of Pt Thin-Films*

Representative SEM measurements of Pt thin films (TF) are displayed in Figure 2a–c. It can be seen that small Pt domains with a size of ca. 5 nm agglomerate to form cauliflower-like structures with a size between 50 nm and 150 nm. The cauliflower morphology is related to a fractal structure. The resulting surface activity and surface area is determined by this structure. The cross-section in Figure 2b shows

the structure of the top-surface layer to be porous and the attaching part to the microporous layer of the gas diffusion electrode to be denser. XRD of Pt TF shows broad diffraction peaks of the indices (111), (200) and (220) crystal structure (see Figure 2d). The average crystallite size estimated by the Scherrer equation is 7.2 nm, which fits to the Pt domain size identified with the electron microscope.

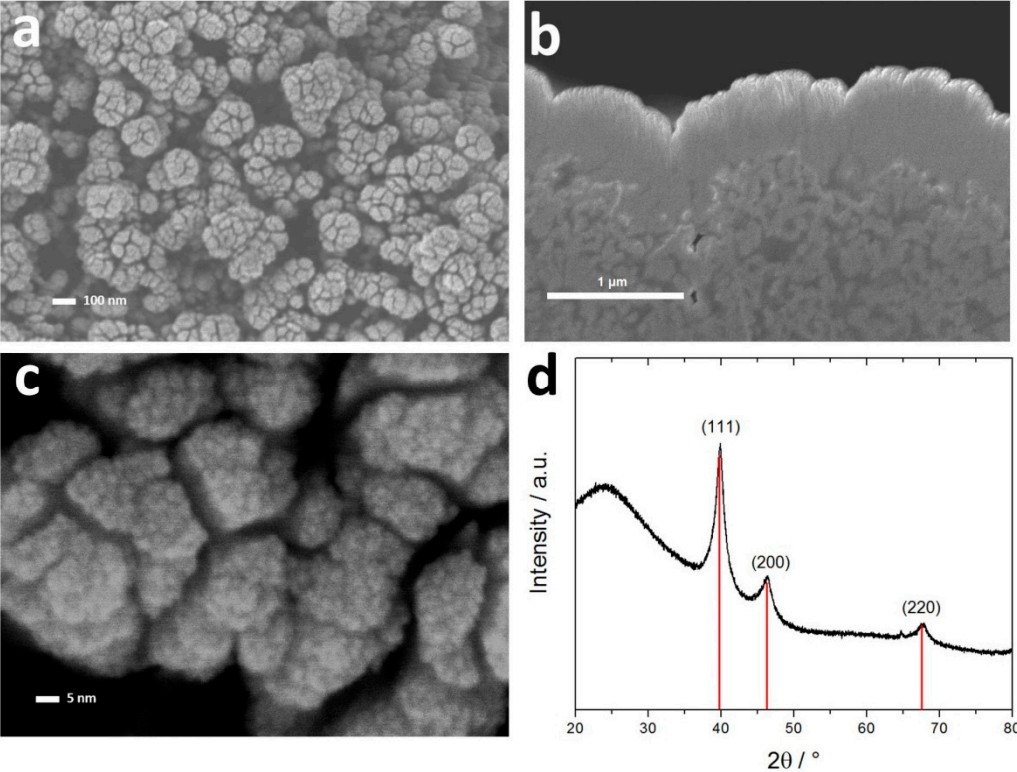

**Figure 2.** (**a**–**c**) SEM micrographs of ~500 nm Pt thin films (TF) on a gas diffusion electrode (GDE) with microporous layer (MPL) H23 C2 from Freudenberg prepared by RF unbalanced magnetron sputtering at 15 W and a working pressure of 5 Pa: top view (**a**), cross-section view (**b**) high-magnification top-view (**c**). (**d**) XRD measurements of Pt TF deposited onto a gas diffusion electrode (Freudenberg, H23 C2) with Pt (111), (200) and (220) diffraction pattern (PDF 00-001-1194).

## 3.2. Electrochemical Characterization

Figure 3 shows the base cyclic voltammograms in argon-saturated electrolyte as well as CO-stripping measurements in order to determine the electrochemically active surface area. Figure 3a displays in detail the cyclic voltammograms of Pt TF with increasing Pt characteristic features for higher loadings, as observed in the hydrogen under potential deposition (0.1–0.4 $V_{RHE}$) region as well as the region where O-containing species are adsorbed (0.7–1.0$V_{RHE}$). The linear sweep voltammogram of the CO oxidation charge transfer is shown in Figure 3b for the different loadings. The sweep shows two CO oxidation peaks, one peak between 0.75 and 0.77 $V_{RHE}$ and a second around 0.82–0.85 $V_{RHE}$, in contrast to the CO oxidation sweep measured in the RDE with one peak at 0.73 $V_{RHE}$. The integrated CO oxidation charge is plotted against the Pt loading with the ideal linear slope crossing 0 and the second lowest loading; see Figure 3c. The oxidation charge increases with loading. However, it can be seen that the CO oxidation charge as function of the loading deviates from an ideal straight line for the highest loading. This could indicate that there is a mass transport problem due to blocking of CO as a reactant or trapped product; however, no CO oxidation peaks were visible in sub sequential sweeps. Thus it can be assumed that the ECSA measurement with CO adsorption is valid.

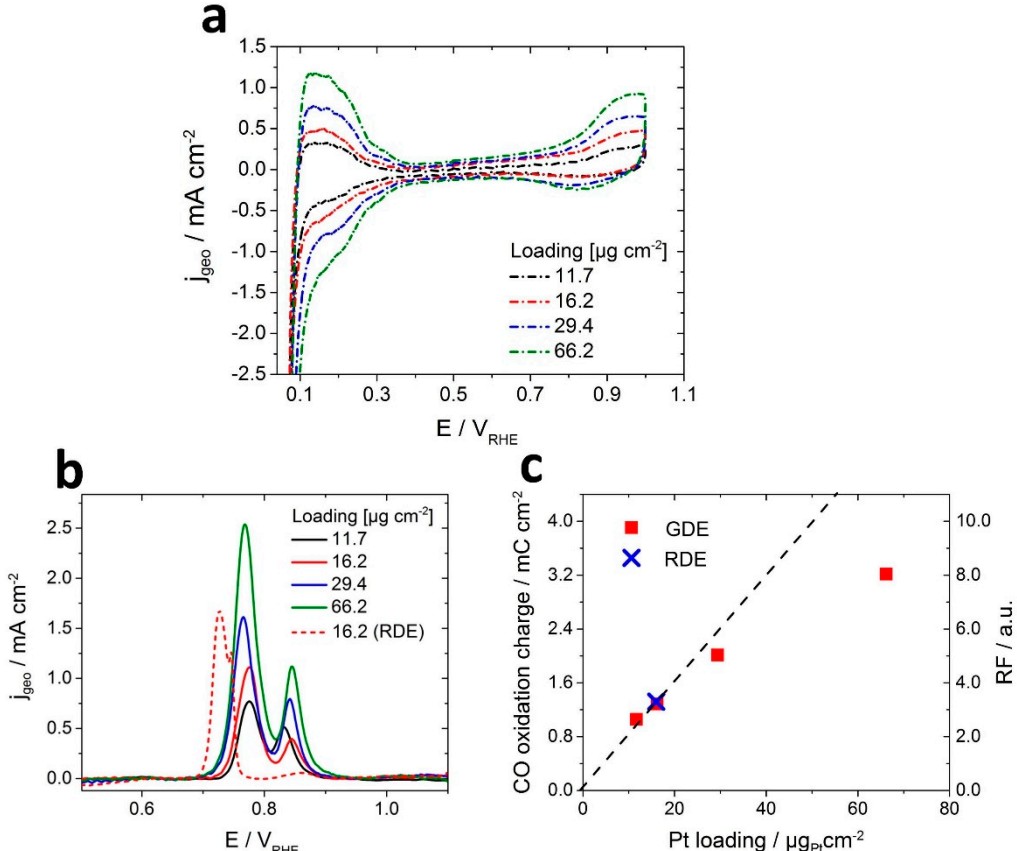

**Figure 3.** (**a**) Loading dependent electrochemical behaviour of Pt TF in GDE setup (298 K, 4 M HClO$_4$, 0% RH), cyclic voltammograms recorded with 50 mV s$^{-1}$ and Ar saturated flow field. (**b**) CO stripping curves recorded with 50 mV s$^{-1}$. (**c**) Plot of CO oxidation charge as function of the Pt loading on the GDE and RDE. The linear slope indicates the ideal correlation between oxidation charge and Pt loading.

Our results from CO-stripping experiments therefore, show that the decrease in ECSA for higher Pt loadings is a result of decreasing Pt utilization. This is supported by the cross-section SEM micrograph displayed in Figure 2b with a higher loading. The graph shows a higher porosity at the near surface region, compared to the region near the Microporous Layer (MPL). Increasing the loading/film thickness leads to lower Pt utilization due to lack of long-range porosity for higher loadings of sputtered Pt thin-films. Consequently, the ECSA measurements have to be carefully done for each loading, especially with loadings above 30 $\mu g_{Pt}$ cm$^{-2}$. The ECSA of the Pt TF GDE is determined to be 20 m$^2$ g$^{-1}_{Pt}$ for the low-loaded 16.2 $\mu g_{Pt}$ cm$^{-2}$ gas diffusion electrodes.

### 3.3. Performance of Pt Thin-Films in the GDE Setup

Figure 4 shows a comparison between the linear sweep voltammograms during the oxygen reduction reaction for both RDE and GDE. As stated in our previous work [28], the GDE enables the measurement of realistic mass transport conditions and high current densities. In contrast, the RDE is mass transport limited due to low solubility of oxygen in the condense phase/liquid electrolyte, but it's well defined hydrodynamics allow extraction of intrinsic kinetic currents [16,17]. Therefore, it is only possible to measure the kinetic current densities at very low overpotentials.

The ORR activity from RDE measurements in acidic electrolyte is typically reported at 0.9 V$_{RHE}$ for less active platinum catalysts and for highly active catalysts it is often only possible to measure at even higher potentials up to 0.95 V$_{RHE}$ [33]. However, fuel cells typically operate at a working potential of in the range of 0.6–0.8 V$_{RHE}$, as indicated in Figure 4. At these potentials, the current in the

RDE is under full mass transport limitation. While in the GDE the current density is not free of mass transport limitations, GDE measurements reveal more realistic current densities [28].

Our GDE setup allows measuring a maximum oxygen reduction current density of ~1 A cm$^{-2}$ using only 16.2 $\mu g_{Pt}$ cm$^{-2}$; this is over two orders of magnitude higher than what is observed in the RDE measurements. The maximum current density is reached at ~0.3 V$_{RHE}$. At potentials below 0.3 V$_{RHE}$ in the GDE it can be seen that the current decreases. This correlates to the region where hydrogen under potential adsorption occurs. Therefore, it is presumed that the decrease in the current density observed between 0.1 V$_{RHE}$ and 0.3 V$_{RHE}$ is due to blocking by protons on the Pt surface which would limit the full oxygen reduction to water. In this region, the catalyst reduces oxygen to hydrogen peroxide, which corresponds to a two-electron transfer compared to a four-electron transfer for the reduction of oxygen to water [34]. It is notable that the maximum current density is reached in the same potential region as the maximum in the extended Koutecky-Levich analysis [23] as well as the potential of zero total charge (PZTC) [35]. The maximum oxygen reduction rate, therefore, correlates with the minimum ion coverage; most likely a competition between protons and oxygen species at the surface and, to a lesser extent, perchlorate anions.

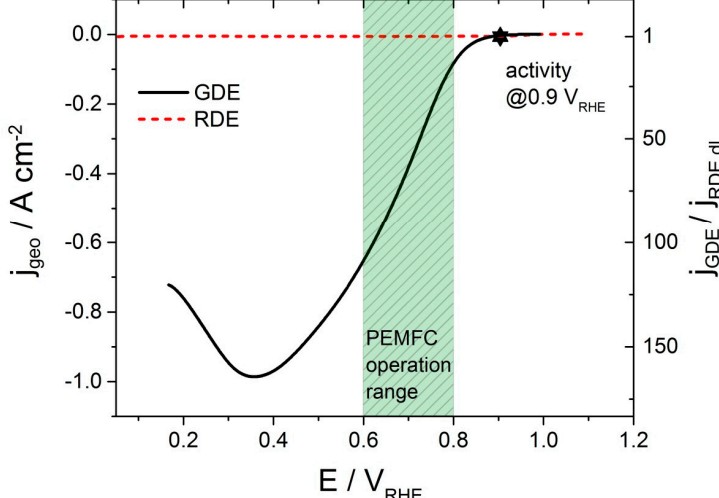

**Figure 4.** Comparison of the current density-potential curves as measured in a GDE setup and a conventional RDE setup. The activity measurement in the GDE setup with oxygen saturated flow field was conducted at 298 K, 4 M HClO$_4$, Nafion 117 membrane, 0% RH and 50 mV s$^{-1}$. The measurement of the activity in the RDE setup was conducted at 1600 RPM, 298 K, 0.1 M HClO$_4$ and 50 mV s$^{-1}$. The catalyst was a Pt thin film prepared by magnetron sputtering with a Pt loading of 16.2 $\mu g_{Pt}$ cm$^{-2}$ for both RDE and GDE. The right axis shows the current density enhancement as compared to the diffusion limited current observed in RDE.

### 3.4. Benchmarking Different Pt Loadings in the GDE Setup

The Pt TF catalyst was tested with different Pt loadings on the GDE, as shown in Figure 5. The obtained linear sweep voltammograms are plotted as kinetic current densities normalized by the catalyst geometric surface area, the specific activities (normalized by the roughness factor) and mass activities. The activities normalized by geometric surface area show the expected increase of current density over the full potential range. At 0.3 V$_{RHE}$, the maximum activity is achieved and reaches 0.75 A cm$^{-2}$ for 11 $\mu g_{Pt}$ cm$^{-2}$ and nearly 1 A cm$^{-2}$ for the highest loading 66 $\mu g_{Pt}$ cm$^{-2}$. The onset potential for the ORR is shifting to lower overpotentials. The specific activities were calculated using the roughness factor determined by CO oxidation; it can be seen that the onset potential stays constant with a value of 0.71 mA cm$^{-2}_{Pt}$ at 0.90 V$_{RHE}$. The specific activity starts to differ in the region from 0.75 V$_{RHE}$–0.8 V$_{RHE}$ for the different loadings. Furthermore, the lowest Pt loading leads to the highest specific activity measured. The Pt TF catalyst with 11 $\mu g_{Pt}$ cm$^{-2}$ achieves a specific activity of 94.0 mA cm$^{-2}_{Pt}$ at 0.65

$V_{RHE}$. While the GDE with the highest loading of 66 $\mu g_{Pt}$ cm$^{-2}$ reaches only a value of 38.6 mA cm$_{Pt}^{-2}$ at 0.65 $V_{RHE}$, corresponding to a "loss" of 59% at 0.65 $V_{RHE}$. The reason for the potential-dependent specific activity is most likely related to mass transport, which can be distinguished due to oxygen transport, proton transport, H$_2$O transport or electronic conductivity. In order to distinguish between the different influences, several operation parameters were changed and discussed later in this paper.

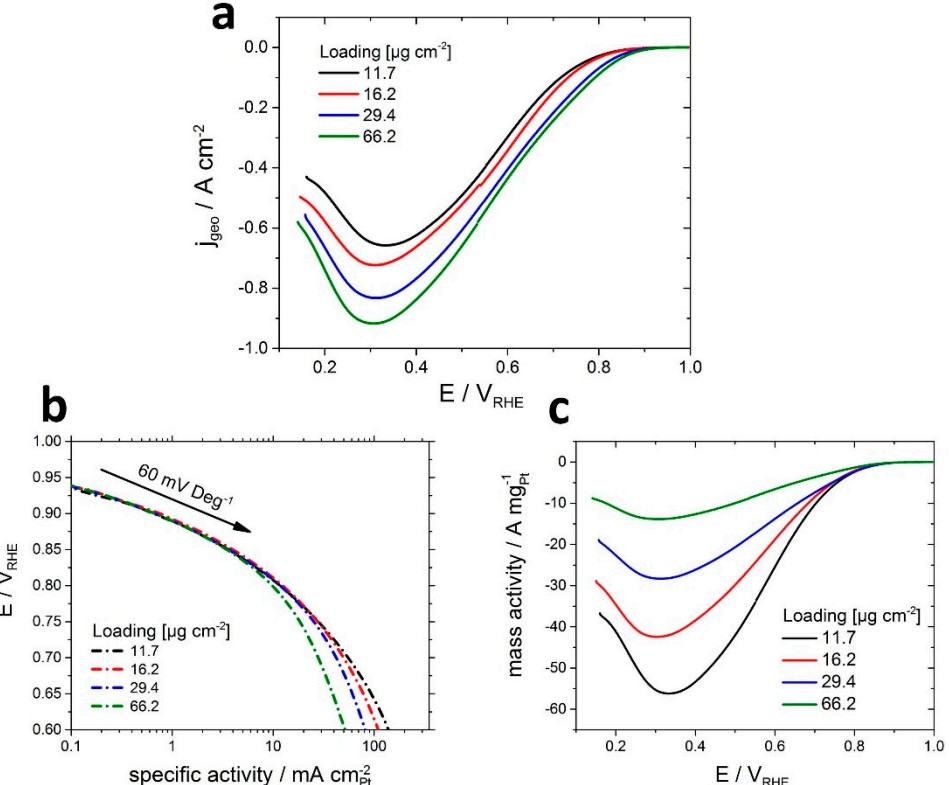

**Figure 5.** Influence of different Pt loadings on the catalyst layer activity (11.7 $\mu g_{Pt}$ cm$^{-2}$, 16.2 $\mu g_{Pt}$ cm$^{-2}$, 29.4 $\mu g_{Pt}$ cm$^{-2}$, 66.4 $\mu g_{Pt}$ cm$^{-2}$). (**a**) Activity normalized by the geometric surface area, (**b**) surface-area specific activity and (**c**) mass specific activity. The measurement of the was conducted in the GDE setup with oxygen saturated flow field (298 K, 4 M HClO$_4$, 0% RH, 100 mV s$^{-1}$).

Comparing the mass activity for the different loadings, the highest activity can again be achieved with the GDE with the lowest loading of 11 $\mu g_{Pt}$ cm$^{-2}$. The activity of the GDE with 11 $\mu g_{Pt}$ cm$^{-2}$ reaches 17 A mg$^{-1}$$_{Pt}$ at 0.65 $V_{RHE}$ and 55 A mg$^{-1}$$_{Pt}$ at 0.3 $V_{RHE}$. The highest loaded GDE achieves only a mass activity of 5 A mg$^{-1}$$_{Pt}$ at 0.65 $V_{RHE}$ and 19 A mg$^{-1}$$_{Pt}$ at 0.3 $V_{RHE}$. At 0.65 $V_{RHE}$ this is a loss of 75% and at 0.3 $V_{RHE}$ the "loss" is 70% of the mass activity compared to the lowest loading.

### 3.5. Systematic Change of Operation Parameters

The Pt TF GDE was evaluated with different operational parameters to understand the activity determining factors of the unsupported catalyst layers (see Figure 6). First, Figure 6a shows the ORR polarization curves of the GDE with a separating membrane, as compared with a non-separated GDE setup with 4 M perchloric acid. The separating membrane increases the geometric activity over the full potential window and at 0.65 $V_{RHE}$ by 104%. To clarify the cause of this drastic increase in activity, we performed a set of measurements with 1 M HClO$_4$ instead of 4 M HClO$_4$ (see Figure 6a). Reducing the molarity of the electrolyte greatly enhanced the activity. Interestingly, the polarization curve for Pt in 1 M HClO$_4$ resembles the one obtained in 4 M HClO$_4$ using a Nafion membrane. This suggests that specific anion adsorption is inhibiting the rate of ORR at high electrolyte concentrations. While perchloric acid is generally considered a non-adsorbing electrolyte, it is known to inhibited ORR

on Pt at high concentrations [36]. Thus, in the absence of a separating membrane, an electrolyte concentration of 1 M $HClO_4$ would be preferable. While no differences were observed in the pure kinetic region (until ~0.8 $V_{RHE}$), the membrane slightly enhanced the activity as compared to 1 M $HClO_4$ in the mass-transport-dominated region (0.3–0.8 $V_{RHE}$), possibly due to enhanced proton transport or reduced flooding of the hydrophilic catalyst layer. At high current densities, oxygen transport is critical. Recently, Kongkanand el. al showed that local $O_2$ resistance in the ionomer film is essential for the performance supported catalysts [4]. While unsupported catalysts are generally free of ionomer, the degree of wetting would influence the oxygen transport in the catalyst layer. It is, therefore, possible that the membrane reduces flooding as a result of less penetrating electrolyte, as compared to measurements without the Nafion membrane. However, from the presented data, it cannot be concluded if adsorbed water on the surface hinders the reaction.

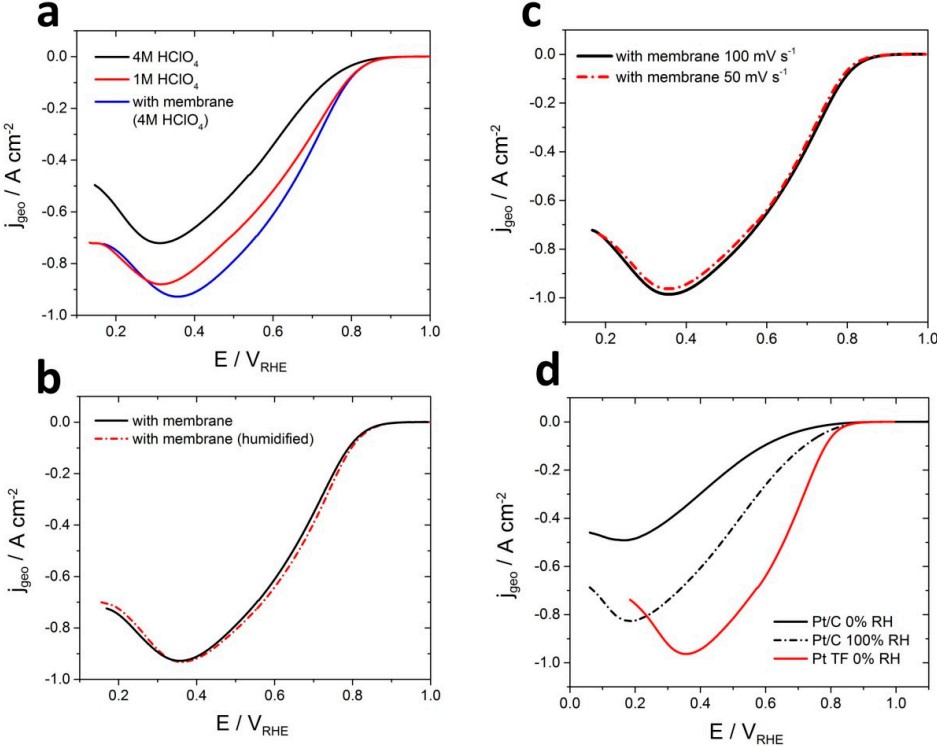

**Figure 6.** Influence of: (**a**) the separating Nafion 117 membrane, (**b**) the humidification on the activity of Pt TF 0% vs 100% relative humidity (RH), (**c**) the scan rate change between 50 mV s$^{-1}$ and 100 mV s$^{-1}$. The measurement of the activity was conducted in the GDE setup with oxygen saturated flow field (16.2 µg$_{Pt}$ cm$^{-2}$, 298 K, 4 M $HClO_4$, 0% RH, 50 mV s$^{-1}$). (**d**) Influence of catalyst system on the catalyst layer performance measured in the GDE with Nafion 117 as membrane in 4 M $HClO_4$ at RT at 50 mV s$^{-1}$ with Pt TF (16.2 µg$_{Pt}$ cm$^{-2}$, in red) and Pt/C (10 µg$_{Pt}$ cm$^{-2}$, in black) [28].

Secondly, the scan rate was varied to 50 mV s$^{-1}$ compared to the 100 mV s$^{-1}$, the results showed negligible decrease in activity (see Figure 6b). This indicates that the current densities are defined by faradaic currents instead of capacitive currents. Capacitive currents increase as the current is proportional to the applied scan rate. Furthermore, it is a sign for a reaction in equilibrium, where mass transport should not be the significant driver limiting the reaction. If mass transport would limit the reaction rate, as the reactant supply drops, the geometric current density should decrease with lower scan rate.

The third operation parameter studied was the humidification of oxygen. It was previously shown that a change of oxygen humidification can have a significant impact on the gained activity of platinum nanoparticles supported on carbon (Pt/C) [28]. However, there is nearly no change in the activity of Pt TF after humidifying the oxygen with 100% relative humidity (RH); see Figure 6c.

This indicates that proton transport is not a limiting step with unsupported Pt catalysts. This is in contrast to Pt/C in GDE, where we observed a doubling in activity with humidification at room temperature, indicating a limiting behaviour; see Figure 6d. In our previous study, Pt/C catalyst was mixed with Nafion [28]. The Nafion ionomer has to be humidified to work as a sufficient proton conductor. In the unsupported Pt TF catalyst, protons seem to be transported on the metallic surface by either Pt-OH or Pt-H. The transport of protons on platinum was first investigated by McBreen et al. [37] and recently, Zenyuk et al. suggested that the transport mechanism might be potential dependent above 0.7 V, where Pt-O and Pt-OH are present on the surface. At potentials below the point of zero charge, the proton is believed to be transported by surface migration [38]. This speculation was already introduced earlier for oxygen reduction catalysts by Debe et al. for Pt on non-conductive organic whiskers [39], the so-called nano-structured thin films (NSTF). This is further supported by the fact that the catalyst layer is also more hydrophobic compared to Pt/C. Thus, at room temperature, sufficient wetting of catalyst layer can be obtained only from water generated by the ORR reaction. Thus, our results support the claim that proton transport on bare Pt is not a limiting step.

The specific activities and ECSAs for the RDE and GDE benchmarking measurements on Pt TFs are summarised in Table 1, as well as compared to commercial Pt/C in the RDE, GDE and MEA. The specific activity of Pt/C in the GDE cell at 60° C and 0.9 $V_{RHE}$ was found to be comparable to the specific activity of the MEA with commercial Pt/C and decreased at room temperature. In general, the specific activity at 0.9 $V_{RHE}$ of GDE and MEA of Pt/C was found to be smaller compared to the RDE measurements. This is also the case for Pt TF where the specific activity in the RDE was benchmarked with 1.82 mA $cm^{-2}_{Pt}$ at 0.9 $V_{RHE}$ and in the GDE with 0.72 mA $cm^{-2}_{Pt}$ with membrane or 0.71 mA $cm^{-2}_{Pt}$ without membrane. The decrease in specific activity for commercial Pt/C is nearly 50% compared to a loss approximately 60% for Pt TF. In the higher current density region, at 0.65 $V_{RHE}$, the specific activity with Pt TF is significantly higher than with Pt/C. In contrast, the current density at 0.65 $V_{RHE}$ by mass is only slightly increased for Pt TF - with 24.1 A/mg$_{Pt}$ for Pt/C and 28.6 A/mg$_{Pt}$ for Pt TF. This indicates that unsupported Pt catalysts have an advantage in activity compared to commercial Pt/C catalysts, especially without humidification at low temperatures. In contrast, at higher loadings, where most devices operate, the low Pt utilization inhibits the performance of the unsupported Pt catalysts. The investigations show that with the help of simple benchmarking of different catalyst systems in the GDE, new ways of developing possible applications can be explored. It turns out that different catalyst systems also require different environmental conditions and can play to their strengths under different operating conditions.

**Table 1.** Comparison of the characterization under different conditions for sputtered Pt thin-film catalyst in GDE and RDE setup and as well as a commercial 46.5 wt % Pt/C catalyst (TEC10E50E, Tanaka) in RDE, GDE and MEA measurements. SA, ECSA and RH stands for ORR specific activity, electrochemical active surface area and relative humidity, respectively.

| Catalyst Layer (Reference) | Electrolyte | Loading [μg$_{Pt}$ cm$^{-2}$] | Temperature/Humidity [°C] | SA @0.9V$_{RHE}$ [mA cm$^{-2}_{Pt}$] | SA @0.65V$_{RHE}$ [mA cm$^{-2}_{Pt}$] | ECSA [m$^2$ g$^{-1}_{Pt}$] |
|---|---|---|---|---|---|---|
| Pt TF RDE (this work) | 0.1 M HClO$_4$ | 16 | rt | 1.82 | - | 21 |
| Pt TF GDE (this work) | Nafion 117/4 M HClO$_4$ | 16 | rt/0% RH | 0.72 | 144 | 20 |
| Pt TF GDE (this work) | 4 M HClO$_4$ | 16 | rt/0% RH | 0.71 | 67.5 | 20 |
| Pt/C RDE [40] | 0.1 M HClO$_4$ | 14 | rt | 0.49 | - | 76 |
| Pt/C GDE [28] | Nafion 117/4 M HClO$_4$ | 5 | rt/100% RH | 0.14 | 25.9 | 93 |
| Pt/C GDE [28] | Nafion 117/4 M HClO$_4$ | 5 | 60/100% RH | 0.18 | 47.9 | 81 |
| Pt/C MEA [41] | Nafion 117 | 90 | 80/100% RH | 0.21 | - | 80 |

## 4. Conclusions

In this paper, we present the electrochemical characterization of platinum thin-films in a GDE setup. Unlike the commonly used RDE method, this half-cell method facilitates investigations under triple phase boundary conditions. Moreover, the GDE setup allows achieving current density values comparable to operating current densities in actual PEMFCs.

Herein, we investigate sputtered Pt with very low loadings, achieving high current densities (close to 1 A cm$^{-2}$) in GDE measurements. Furthermore, we show that different parameters such as loading, electrolyte concentration, humidification and Nafion membrane influence the performance of the Pt catalyst layer. By carefully measuring the activity with different loadings in the GDE, we observe that specific activities at potentials down to 0.75–0.80 V$_{RHE}$ can be determined independently of the applied loading. The values obtained in the GDE were two-and-half times lower than those measured with the RDE technique using the same catalyst. Our results confirm that activity values obtained in liquid half-cells are not directly transferable to real devices. This is mainly due to transport resistance at the triple phase interface and not a result of higher loadings or different testing parameters.

Interestingly, our results show that at fuel-cell relevant overpotentials, the activity is highly dependent on the loading and use of the membrane. Furthermore, we observed that humidification had no influence on the activity of catalyst layer for the unsupported catalyst. This is in contrast to Pt/C, where the use of humidification nearly doubled the catalytic activity. These findings highlight the potential impact of GDE half-cell benchmarking. While RDE remains a valuable tool for observing trends in intrinsic catalytic performance, the properties of the catalyst layer have to be investigated under more realistic conditions. Using sputtered platinum as a model catalyst, we show that the catalyst layer can be efficiently evaluated in the fuel cell operation region using a GDE half-cell setup. It is our hope that this enables accelerated development of fuel cell catalysts by shortening expensive and time-consuming MEA optimization.

**Author Contributions:** Conceptualization, A.W.J. and G.S.; formal analysis, A.W.J. and G.S.; investigation, A.W.J. and G.S.; data curation, A.W.J. and G.S.; project administration, M.A. and M.E.-E.; resources, V.B.; supervision, M.E.-E. and M.A.; writing—original draft preparation, A.W.J. and G.S.; writing—review and editing, M.A. and M.E.-E.

**Funding:** This research was funded by the Danish Innovation Fund (4 M center) and the Danish 350 DFF through (grant number 4184-00332). M.E.-E. and A.W.J. gratefully acknowledge the Villum Foundation V-SUSTAIN grant 9455 to the Villum Center for the Science of Sustainable Fuels and Chemicals.

**Conflicts of Interest:** The authors declare no conflict of interest.

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
