# Peer review of "Sputtered Platinum Thin-films for Oxygen Reduction in Gas Diffusion Electrodes: A Model System for Studies under Realistic Reaction Conditions"

_surfaces, doi:10.3390/surfaces2020025_

Round 1

Reviewer 1 Report

The paper extends to unsupported Pt thin films the use of GDE half cells for the assessment of catalyst performance in ORR reaction. The experiments are designed to test different reaction parameters such as humidification, effect of the nafion membrane and Pt loading. The conclusions are fairly supported by the data, but the effect of Pt loading on the morphology is not thoroughly documented. So the hint that "decreasing Pt utilization" could be the cause of decrease in ECSA as a function of Pt loading remains conjectural, even if reasonable. To this concern, the only SEM and XRD data reported are not attributed to a specific sample and therefore do not allow a comparison of morpohlogy among the different Pt loadings. The cauliflower-like morphology seems to be related a fractal structure of the films, that could affect the reactivity, as extensively reported in the literature. 

The step in the XRD patterns is likely 0.05 and not 0.5 2theta, the anode target of the diffractometer is not specified. 

In conclusion, the paper is worth of publication in Surfaces. In my opinion, a more detailed study of the correlations between structure/morphology and catalytic properties could improve the paper.

Author Response

Comment 1.1

“The paper extends to unsupported Pt thin films the use of GDE half cells for the assessment of catalyst performance in ORR reaction. The experiments are designed to test different reaction parameters such as humidification, effect of the nafion membrane and Pt loading. The conclusions are fairly supported by the data, but the effect of Pt loading on the morphology is not thoroughly documented. So the hint that "decreasing Pt utilization" could be the cause of decrease in ECSA as a function of Pt loading remains conjectural, even if reasonable. To this concern, the only SEM and XRD data reported are not attributed to a specific sample and therefore do not allow a comparison of morpohlogy among the different Pt loadings. The cauliflower-like morphology seems to be related a fractal structure of the films, that could affect the reactivity, as extensively reported in the literature.” 

Response 1.1

The decreasing Pt-utilization is based on the CO-stripping experiments. To support this, we also performed physical characterization by SEM on a ~500 nm sample, which is shown figure 2. In the SEM micrographs, a considerable higher porosity is observed at the surface compared to the bulk. This observation agrees with the decrease in ECSA as a function loading measured in the CO-stripping experiments. Concerning the cauliflower morphology, we agree it is related to a fractal structure and this can affect reactivity both in terms of specific activity and ECSA. The cauliflower structure increases the amount of defects, which might increase activity as well.

Action 1.1

We now clearly specify that we could not attribute the morphology to specific samples and that the decrease in Pt utilization is mainly concluded by the observed decreasing in the CO adsorption data. We have added the following sentence:

“Our results from CO-stripping experiments therefore show that the decrease in ECSA for higher Pt loadings is a result of decreasing Pt utilization. This is supported by the cross-section SEM micrograph displayed in Figure 2b with a higher loading”

In relation to the morphology, we have added the following sentence:

“The cauliflower morphology is related to a fractal structure. The resulting surface activity and surface area is determined by this structure.”

Comment 1.2

“The step in the XRD patterns is likely 0.05 and not 0.5 2theta, the anode target of the diffractometer is not specified.” 

We thank the reviewer for noting this and have made the appropriate corrections.

Action 1.2

In the experimental section, we now added Cu as the anode material and corrected the step width to 0.05°.

Comment 1.3

“In conclusion, the paper is worth of publication in Surfaces. In my opinion, a more detailed study of the correlations between structure/morphology and catalytic properties could improve the paper.”

We thank the reviewer for the positive comments about our work. We would like to note that the main purpose of this work was to evaluate the effects of different parameters in gas diffusion electrodes, rather than evaluating the structure-activity relationships of the sputtered catalyst. For this purpose, we utilized the unsupported Pt catalyst as a model system. The benefit of the sputtered catalyst system is that it enabled precise control of catalyst loading and a high degree of sample uniformity.  

Reviewer 2 Report

The study addresses an important issue for future energy systems, the development of catalysts for the oxygen reduction reaction in fuel cells. An experimental setup using a gas diffusion electrode is introduced. Compared to rotating disc electrodes, this setup has the advantage to be able to operate under conditions closer to those in real fuel cells. Compared to the catalyst testing in fuel cells, the GDE setup uses much less expensive material, providing substantial advantage for reserach.

The study is well presented. No issues are detected. Publication is recommended in the present form.

Author Response

We appreciate the positive comments on our manuscript.

Reviewer 3 Report

Sputtered platinum thin-films for oxygen reductionin gas diffusion electrodes – a model system forstudies under realistic reaction conditions “\

The manuscript is well organized and the below comments should be applied. The presented manuscript should be revised.

1-   Abstract should be revised and some important obtained data should be presented

2-   Introduction section should be more completed with some published research for this new test system

3-   For comparison between RDE and MEA, some references should be added.

4-   Fig. 2b, the quality is low. The particle size distribution also could be added. EDX mapping, also is could be helpful to see the distribution of Pt on GDE.

5-   HRTEM results can be interesting for this work.

6-   After Pt deposition, how much the porosity of GDE was changed?

7-   Is the GDE teflonized? How is the hydrophilicity-hydrophobicity of the GDE?

8-   The plain GDE did have any activity in this setup?

9-   Fig. 6b: How the authors did humidify the gas? How much was the humidity level?

10-The results of with membrane and without membrane should be more discussed with more details.

11-The proton transfer in unsupported Pt DGE should be more discussed. Some evidences.

12-How much is the thickness of Pt film? Did it penetrated in the GDE? 

13-How is the durability of this Unsupported Pt? a test should be done to show the stability of this catalyst.

14-Did the authors any real fuel cell test (MEA) for this type of Pt-GDE?

Author Response

Comment 3.1

“Abstract should be revised and some important obtained data should be presented”

Response 3.1

We have revised the abstract; now it includes more details on the key findings.

Action 3.1

We have added the following sentence:

The specific activity could be measured independent of the applied loading at potentials down to 0.80 VRHE reaching a value of 0.72 mA cm-2 at 0.9 VRHE in the GDE

Comment 3.2

“Introduction section should be more completed with some published research for this new test system.”

Response 3.2

We thank the review for this comment and acknowledge that the introduction section was not comprehensive in relation to recent studies.

Action 3.2

We added a paragraph to the introduction highlighting recent advances in with gas diffusion electrodes in more detail:

“These works include measurements in concentrated phosphoric acid at elevated temperature [25], methanol and ethanol oxidation [26] and Pt/C in HClO4 showing comparable activity to MEA, both at room temperature [27] and at elevated temperatures using a Nafion separating membrane [28].”

Comment 3.3

“For comparison between RDE and MEA, some references should be added.”

Response 3.3

We thank the reviewer for emphasizing this. We had included the appropriate references for comparison between RDE and MEA in table 1. However, we understand they were not easily seen.

Action 3.3

We revised table 1, so the relevant references are more clearly displayed (they can now be found in the first column).

Comment 3.4

“Fig. 2b, the quality is low. The particle size distribution also could be added. EDX mapping, also is could be helpful to see the distribution of Pt on GDE.”

Response 3.4

Unlike application with catalyst inks by brushing or spraying, the sputtering allows to apply the catalyst directly on the MPL surface (see response 3.12). As a result, we have not performed any EDX mapping of the GDE. As these catalysts are sputtered thin films with extended nanostructures formed by fractal growth instead of independent nanoparticles, it is not possible to have a particle size distribution.

Action 3.4

We have updated fig 2b, with a SEM micrograph from the same spot without the “blurry” background. Furthermore, we have now corrected our own definition; we have renamed the structures to Pt domains (instead of Pt particles).

Comment 3.5

“HRTEM results can be interesting for this work.”

Response 3.5

We agree that high resolution TEM could by interesting, and we consider including this in future works.

Comment 3.6

“After Pt deposition, how much the porosity of GDE was changed?”

Response 3.6

We assume that the porosity in the bulk of GDE didn´t change, as the sputtering process is highly surface sensitive (see response 3.12). However, the sputtered Pt film is denser than the MPL layer, as observed in figure 2.

Comment 3.7

“Is the GDE teflonized? How is the hydrophilicity-hydrophobicity of the GDE?”

Response 3.7

According to the manufacturer, there is no hydrophobic treatment on fibrous part of the GDE. (https://fuelcellcomponents.freudenberg-pm.com/-/media/Files/fuelcellcomponents,-d-,freudenbergpm,-d-,com/FPM_technical_data_sheet_gdl_ENG_2018-07-04.pdf). However, the microporous layer (MPL), have been teflonized, as also confirmed experimentally in a recent study (10.1002/fuce.201700181).  

Comment 3.8

“The plain GDE did have any activity in this setup?”

Response 3.8

In the applied potential window, there was no measurable activity observed for the GDE without sputtered catalyst.

Comment 3.9

“Fig. 6b: How the authors did humidify the gas? How much was the humidity level?”

Response 3.9

The inlet gas was humidified in a gas bubbling humidifier, as stated in the Materials and Method section. The humidification in fig. 6b was 0 and 100 % relative humidity (RF), for clarity this is now included in the figure caption.

Action 3.9

We revised the figure caption in fig. 6b, now it clearly displays the humidity level:

(b) the humidification on the activity of Pt TF 0% vs 100% relative humidity (RH)”

Comment 3.10

“The results of with membrane and without membrane should be more discussed with more details.”

Response 3.10

We thank the reviewer for this comment and acknowledge the need for a more detailed discussion on the effect of the membrane.

Action 3.10

We now added a discussion on the effect of the membrane, mainly the effect on the oxygen transport is discussed, now in more specific manner, in the following paragraph:

“At high current densities, oxygen transport is critical. Recently, Kongkanand el. al showed that local O2 resistance in the ionomer film is essential for the performance of supported catalysts [4]. While unsupported catalysts are generally free of ionomer, the degree of wetting would influence the oxygen transport in the catalyst layer. It is therefore possible that the membrane reduces flooding as a result of less penetrating electrolyte, as compared to measurements without the Nafion membrane. However, from the presented data, it cannot be concluded if adsorbed water on the surface hinders the reaction.”

Comment 3.11

“The proton transfer in unsupported Pt DGE should be more discussed. Some evidences.”

Response 3.11

This poses an interesting question. Based on previous literature, it is possible to get some insights on the mechanism of proton transport on bare Pt. However, from our perspective, the real mechanism of proton transport is still in question and should be investigated in more detail.

Action 3.11

We added the following to the discussion on the proton transfer mechanism:

“The transport of protons on platinum was first investigated by McBreen et al. [37] and recently Zenyuk et al. suggested that the transport mechanism might be potential dependent above 0.7 V, where Pt-O and Pt-OH are present on the surface. At potentials below the point of zero charge, the proton is believed to be transported by surface migration [38]

Comment 3.12

“How much is the thickness of Pt film? Did it penetrated in the GDE?”

Response 3.12

The Pt did not penetrate in the GDE as the sputtering technology is only covering the top surface. In fact, it is typically demanding to sputter-coat the corners and profiles sufficiently, which leads to problems with highly structured work pieces. We calculated the thickness of Pt catalyst layers by the nominal thickness measured by a profilometer with higher loadings.

Action 3.12

We have now added the following sentence to the Materials and Method section:

“A profilometer (Dektak 3ST) was used to measure the film thickness. Corresponding to the Pt loadings of 11.7 µg cm-2, 16.2 µg cm-2 , 29.4 µg cm-2 and 66.2 µg cm-2, the films had a thickness of 15 nm, 20 nm, 37 nm and 83 nm, respectively”

Comment 3.13

“How is the durability of this Unsupported Pt? a test should be done to show the stability of this catalyst.”

Response 3.13

We would like to thank the reviewer for pointing this out. For the current study, we did not perform any stability test on this catalyst, as the performance at higher loadings was not ideal. The main goal of this study was to evaluate the unsupported catalyst as a model system in order to conclude the effects of different parameters under realistic conditions.

Our results suggest that this GDE half-cell setup allows to bridge the gap and give the experimentalists a fast approach realistic catalyst benchmarking. In addition, stability tests performed in GDE setup might provide a more realistic picture compared to the current RDE method, which often fails to predict the actual losses observed in real fuel cells. We thus plan to do systematic benchmarking studies on the stability of Pt-based catalysts by means of GDE in the future.

Comment 3.14

“Did the authors any real fuel cell test (MEA) for this type of Pt-GDE?”

Response 3.14

No, unfortunately, we did not have the opportunity to do any test at MEA level yet. We agree these studies should be done in the future. In fact, we plan to use this method as a straight-forward technique for catalyst layer optimization prior to actual fuel cell testing.

Round 2

Reviewer 3 Report

The revised manuscript has been improved and it can be accepted.